# A Dynamic Algorithm for Interference Management in D2D-Enabled Heterogeneous Cellular Networks: Modeling and Analysis

**DOI:** 10.3390/s22031063

**Published:** 2022-01-29

**Authors:** Md Kamruzzaman, Nurul I. Sarkar, Jairo Gutierrez

**Affiliations:** Department of Computer Science and Software Engineering, Auckland University of Technology, Auckland 1010, New Zealand; zaman.kamruzzaman@aut.ac.nz (M.K.); jairo.gutierrez@aut.ac.nz (J.G.)

**Keywords:** Device-to-Device (D2D) communication, heterogeneous networks, HetCNets, interference management

## Abstract

To supporting a wider and diverse range of applications, device-to-device (D2D) communication is a key enabler in heterogeneous cellular networks (HetCNets). It plays an important role in fulfilling the performance and quality of service (QoS) requirements for 5G networks and beyond. D2D-enabled cellular networks enable user equipment (UE) to communicate directly, without any or with a partial association with base stations (eNBs). Interference management is one of the critical and complex issues in D2D-enabled HetCNets. Despite the wide adoption of D2D communications, there are very few researchers addressing the problems of mode selection (MS), as well as resource allocation for mutual interference in three-tier cellular networks. In this paper, we first identify and analyze three key factors, namely outage probability, signal-to-interference and noise ratio (SINR), and cell density that influence the performance of D2D-enabled HetCNets. We then propose a dynamic algorithm based on a distance-based approach to minimize the interference and to guarantee QoS for both cellular and D2D communication links. Results obtained show that outage probability is improved by 35% and 49% in eNB and SCeNB links, respectively, when compared with traditional neighbor-based methods. The findings reported in this paper provide some insights into interference management in D2D communications that can help network researchers and engineers contribute to further developing next-generation cellular networks.

## 1. Introduction

In cellular networks, device-to-device (D2D) communication is an emerging technology in which two nearby user’s equipment communicate with each other without any base station (BS) or core network support. Due to the short communication range between a D2D pair, D2D communication provides several advantages in terms of spectrum efficiency, throughput, latency, power management, coverage expansion, and capacity improvement by reusing radio resources. Furthermore, D2D communication enables new services such as public safety, location-based commercial proximity; content sharing of files, videos or pictures; gaming, connectivity extension, and traffic offloading [1]. Owing to these benefits in 5G networks and beyond, D2D communication is a key enabler technology [1].

The growing popularity of high-end user devices and the diversified content of mobile multimedia has contributed, during the last decade, to an exponential growth in both mobile broadband traffic and end-user demand for faster data access. In addition, the number of mobile devices and connections are growing exponentially; by 2022, there will be 12.3 billion mobile devices and connections compared to 8.6 billion in 2017 [2]. As per the recently published Cisco visual networking index [2], mobile data traffic has increased 18-fold, from 400 petabytes to 7.2 exabytes per month, from 2011–2016, with further tenfold growth expected, reaching 77 exabyte per month by 2022. Moreover, with enhanced mobile broadband (eMBB), ultra-reliable and low latency communication (URLLC), and massive machine type communications (mMTC), different services are surely forthcoming for 5G networks and beyond [3]. Managing such a high user density and the resulting immense data volumes is a major concern for cellular network operators [1].

Furthermore, in the 3rd Generation Partnership Project (3GPP) Release 12 for proximity-based services (ProSe) and group communication system enablers (GCSE), D2D communications is an integrated module in the Long-Term Evolution Advanced (LTE-A) standard [1]. During natural disasters such as earthquakes or hurricanes, a replacement for the traditional network can be set up quickly with the help of D2D functionality. In addition, multi-hop cooperation between devices can help to enhance coverage, since at those times, D2D may be the only mode of communication in no coverage areas. Hence, D2D communication in cellular networks will bring significant performance gains in terms of data offload (due to direct communications), improved spectrum efficiency (due to reuse cellular resources), coverage extension (by providing improved connectivity among UEs), and content sharing/dissemination [4,5,6].

However, to maximize the benefits of D2D communications, there are many open challenges that need to be thoroughly addressed [6]. These challenges include mode selection, neighbor discovery, interference and radio resource management, energy consumption, coexistence of D2D with small cells, mobility management, network security, etc. Among them, interference management (IM) in a heterogeneous scenario comparing all tiers simultaneously is very important and complex [1].

Figure 1 shows possible interference scenarios in a three-tier cellular network where eNB, SCeNB, and D2D pairs will reuse cellular resources to communicate simultaneously and introduce mutual interference among different tiers. Hence, to achieve the benefits of D2D communication in cellular networks, it is essential to manage interference, and by selecting an appropriate mode of transmission, power control, resource allocation, antenna systems, and location restriction, we can achieve this. Based on spectrum usage, the main two categories of D2D communications are: (1) D2D overlay, where a dedicated orthogonal spectrum is used for D2D communications within conventional cellular users in a cell [7]; and (2) D2D underlay, where conventional cellular spectrum will be shared with D2D communications, which leads to better spectrum utilization at the cost of complex interference scenarios [3].

In this paper, we identify various key factors that contribute to interference in an underlay heterogeneous cellular network where uplink (UL) resources are shared among D2D pairs and small cells. The main contribution and strength of this paper is the emphasis on the fact that a dynamic algorithm is required to handle interference management in D2D-enabled heterogeneous cellular networks.

The main contributions of the paper are as follows.

We propose a dynamic algorithm called Acceptance Interference Region (AIR) to provide a solution to the problem of guaranteeing a strict QoS for all links in D2D-enabled heterogeneous cellular networks. A distance-based approach is used to achieve guaranteed link quality. The proof of AIR is provided in Appendix A;We propose an efficient ON/OFF algorithm to provide a solution to the problem of achieving maximum transmission capacity in the network;We develop a mathematical model containing the network, SINR, and small cell density models for system performance modeling and analysis. To this end, we derive the outage probabilities of D2D links, macro-cell links, and small cell links for system performance analysis. We also provide analysis and proof (see Appendix B) to show how small cell density and the number of D2D pairs affect the communication link quality. We validate our analytical models using a MATLAB-based simulation.

The rest of this paper is organized as follows. Section 2 reviews the literature on interference management in D2D-enabled heterogeneous cellular networks. Section 3 presents the system model, including network, signal-to-interference-plus-noise ratio (SINR), and small cell density models. The proposed AIR dynamic algorithm is presented in this section. Section 3 also presents theoretical analysis covering coverage probability and spectral efficiency. The system performance is evaluated in Section 4. The simulation results are also presented in this section. Conclusions are drawn in Section 5. Table 1 lists the key mathematical notations and abbreviations used in this paper.

## 2. Related Work

In a D2D-enabled heterogeneous cellular network, state-of-the-art researchers have proposed different methods, working principles and techniques to manage interference. Based on interference behavior, all of these can be categorized as interference avoidance, interference cancellation, and interference reduction.

In interference avoidance techniques, DUEs (D2D user equipment) are not allowed to use the cellular resources up to a certain area of coverage that is defined by a predefined signal-to-interference-noise ratio (SINR); the interference limited area is normally derived with a specific transmission power level at the UEs [8] or based on the distance as well as channel quality of the UEs [9]. In [3], to make a robust cellular link and to reduce interference between the cellular and D2D layers, an interference aware power allocation (IAPA) solution is proposed to improve spectral efficiency, throughput, and outage probability of cellular links. An analytical framework for MS is developed for two-tier cellular networks by using Markov-chain theory in [10] for a single user scenario, and joint MS, resource allocation, and scheduling optimization is formulated using a greedy heuristic algorithm. In [11], a deep reinforcement learning-based dynamic spectrum access scheme for D2D communication in heterogeneous cellular network is proposed using the location of CUEs (cellular user equipment) where link QoS is guaranteed. In [12], the authors first derived the outage probabilities of D2D, macro-cells, and small cell links, and then presented the lower distance boundaries from the D2D transmitter to the macro-base station (eNB) and small cell receiver, and from the base station receiver to a D2D receiver. Finally, an optimal small cell deployment density that ensures quality of service (QoS) requirements for both D2D and cellular communications is proposed. In [13], the authors proposed an interference limited area (ILA)-based D2D management scheme where an appropriate power control algorithm is used for mitigating interference. In their work, multiple DUEs, but a single macro-cell are considered, and DUEs can only initiate their communication outside a restricted area. As in [12], lower distance boundaries between the D2D receiver and cellular cells (both macro- and small cells) are defined, and based on an interference management scheme and upper boundary, a small cell density is proposed in which D2D devices can only reuse cellular resources effectively to meet QoS requirements [14].

In interference cancellation techniques, multiple antenna systems, beam-forming, or pre-coding are used. The major advantage of these approaches is to manage or eliminate the interference without reducing the transmission power. Hence, transmission rate is improved, but at the cost of additional computation power and communication overheads. In [15], an overlay mode of D2D communication in mm-wave 5G networks is proposed in which alternate offer bargaining game theory is used to improve system throughput without compromising SINR.

In interference coordination approaches, CUE and DUE transmit power and channel assignment are optimized to maximize the objectives. However, this often requires centralized computation at the eNB. In addition, to reduce the system complexity, D2D pairs can reuse radio resources from only one CUE, regardless of other available CUEs with better channel conditions. In [16], both line-of-sight (LoS) and non-line-of-sight (NLoS) transmissions are considered in a practical path loss model, and the authors propose mode selection techniques based on the maximum received signal strength for each UE to control D2D-to-cellular interference. To minimize the impact of interference in D2D-enabled cellular networks, in [17], the authors propose a distributed algorithm using matching theory to allocate the appropriate resources. The researchers first modeled the spatial distributions of UEs and BSs in HetCNets using a homogeneous poison point process, and finally proposed a resource allocation algorithm wherein the QoS of D2D communications is guaranteed. Similar to [13], in [18], a joint power control and mode selection scheme is proposed to minimize interference in D2D-enabled HetNet cellular networks in which power control is used to adjust the interference limited area dynamically and mode selection is used to maximize the spectral efficiency. In addition, to improve performance and mitigate in-band emission interference (IEI) for cellular links in D2D-enabled cellular network, Albasry et al. [19] propose a distance-density-based (DDB) frequency resource grouping strategy and optimal power allocation (OPA) algorithm. The DDB strategy is used to give the higher priority in QoS for D2D links, whereas OPA is used when the number of D2D links is more important. Albasry et al. use stochastic geometry and an analytical approach for their modeling in two-tier networks.

In [20], Hassan et al propose a weighted bipartite matching algorithm to minimize interference in a two-tier D2D-enabled cellular network and use local search techniques to further improve their outcomes. Furthermore, Hassan et al. compare their proposed algorithms with a two-phase auction-based fair and interference aware resource allocation algorithm to show the performance of their algorithms. Huynh et al., in [21], propose an interference management algorithm to maximize the performance of D2D communication without compromising QoS requirements for cellular links in both UL and DL by optimizing admission control, power control, and resource allocation. The main disadvantage of their work is that resource sharing with multiple D2D pairs is not considered and their analysis is limited to a two-tier network.

In [22], Chen et al. propose two different resource sharing strategies for co-tier and cross-tier interference separately, along with their respective power control mechanisms. To optimize the problem, here, convex optimization and 0-1 assignment problem techniques are used for power control and resource allocation, respectively. In [23], a concatenated bi-partite matching (CBM) method is proposed to mitigate interference by appropriate sub-band assignment (SA) and resource allocation (RA). In this proposal, user equipment (UE) density, e-node-B (eNB) density, and the switching frequency of small cells are adaptively determined, and the effect of UL power control is managed by full and truncated channel inversion methods. Initially, CBM is developed on single matching, which is eventually generalized to multiple cells for SA and RA DUEs. By using Monte Carlo methods, an iterative scheme is proposed wherein mode selection and power control are used jointly. The interference on each RB (resource block) is measured and the transmission power is adjusted to achieve the targeted throughput without affecting the CUE performance.

In [24], the proposed scheme is only validated using a single cell with different radii. State-of-art-researchers proposed another smart mode and power selection-based distribution approach in [25], in which real-time information of local traffic channels and surrounding node information is used. Here, dynamic switching is adopted to control interference; otherwise, communication continues using the cellular mode. Moreover, to resolve the issues arising from spectrum sharing in D2D-enabled cellular networks, a location-related strategy for mode selection and a spectrum sharing algorithm are proposed in [26], in which devices form a coalition to share spectrum among DUEs and CUEs.

Most of the aforementioned state-of-the-art techniques focus on interference management in single or two-tier cellular networks. Moreover, for interference management, most researchers consider only one D2D pair and the effects of macro-cells and multiple small cells are neglected, and very few papers focus on DUEs in a D2D pairs that are attached to different eNBs, or with one in an eNB and another in a small cell. In 5G and beyond, ultra-dense networks (UDNs) will be deployed and in practice, multiple CUEs may associate with different cellular layers. Therefore, additional D2D layers will create the most prominent challenge to minimizing interference. Despite the prominent usage of D2D communications, there are very few current researchers who are working to address the MS and resource allocation for mutual interference problems in three-tier cellular networks. Furthermore, none of the authors mentioned here reviewed the worst and, at the same time, the most challenging interference case, which is when all three-tiers of the networks (i.e., eNB, SCeNB, and D2D communication) mutually interfere [1] and the effect of multiple SCeNBs and D2D pairs are considered. This is the main difference between the existing approaches and our proposal for managing interference in a three-tier heterogeneous cellular network.

## 3. System Model

### 3.1. Network Model

For modeling a three-tier D2D-enabled heterogeneous cellular network, we considered a macro-cell (eNB) at the center of the coverage area with radius R, which is surrounded by several small cells (SCeNBs) and D2D pairs. Small cells are randomly distributed within the macro-cell (eNB) coverage area. Due to the random and unpredictable location of small cells, the spatial position of the small cells is modeled by using a homogeneous PPP $\phi_{s}$ with density $\lambda_{s}$, and DUEs are also distributed in the network region according to another independent homogeneous PPP $\phi_{d}$ with density $\lambda_{d}$. Here, DUEs, SCeNB, and evenly distributed CUEs are denoted by $j\in \{1,2,\ldots ,N_{D}\}$, $k\in \{1,2,\ldots ,N_{S}\}$, and $i\in \{1,2,\ldots ,N_{C}\}$, respectively. Moreover, for modeling large scale wireless networks and capturing the effects of network topology on network performance, stochastic geometry is more suitable [27]. In a three-tier network as shown in Figure 1, each UE can communicate in any one of the following modes: (1) DUEs can communicate directly without base stations using the D2D communication mode; (2) CUEs can communicate with each other through the eNB; this is known as the macro-cell or cellular communication mode; and (3) SUEs can communicate with each other through the SCeNB; this is known as the small cell communication mode. To mitigate the intra-cell interference (between UEs within the same cell), cellular resources are assigned orthogonally and each cellular UE uses separate RBs. Co-channel interference can be limited by allowing only one D2D link to share the resources of a cellular link at a time.

We assume all channel gains are independent of each other, independent of the spatial locations, symmetric, and identically distributed (i.i.d.). For simplicity of analysis, only a Rayleigh fading environment is considered and channel coefficients are assumed to be exponentially distributed. In such D2D-enabled HetCNets, the channel model is composed of large-scale path loss and small Rayleigh fading, so in general the received signal can be expressed as [14]:(1)$$P_{r}=P_{t}h_{xy}D^{-\alpha }$$
where $P_{t}$ is the transmission power, α is the path loss exponent, *D* is the distance between the transmitter x and the receiver y, and $h_{xy}$ is the channel coefficient for that particular link.

A receiver can decode a message successfully if and only if the SINR at the receiver is greater than a specific threshold $\gamma_{th}$. If the SINR at the receiver does not meet $\gamma_{th}$, the link experiences an outage. Thus, the outage probability of the x,y link can be expressed as:(2)$$P_{out}=P_{r}\left\{\gamma_{y}\leq \gamma_{th}\right\}$$
where $P_{r}\left(.\right)$ is the outage probability for a minimum SNIR threshold $\gamma_{th}$.

Let us consider *e*, *s*, *i*, *j*, and *k* subscripts to denote the serving eNB, the serving SCeNB, the *i*th CUE, the *j*th D2D pairs, and the *k*th SUE, respectively. The subscripts *t* and *r* denote the transmitter and the receiver of the D2D pair, respectively. In the context of the above defined network where UL cellular resources are shared by D2D pairs and small cells, the mutual interference at different receiver can be expressed as:(3)$$I_{i}=\sum_{j=1}^{N_{D}}P_{j}h_{j,e}d_{j,e}^{-\alpha }+\sum_{k=1}^{N_{S}}P_{k}h_{k,e}d_{k,e}^{-\alpha }+N_{0}$$
(4)$$I_{j}=P_{i}h_{i,r}d_{i,r}^{-\alpha }+\;\sum_{j^{\prime }=1,j^{\prime }\neq j}^{N_{D}}P_{j^{\prime }}h_{j^{\prime },r}d_{j^{\prime },r}^{-\alpha }+\sum_{k=1}^{N_{S}}P_{k}h_{k,r}d_{k,r}^{-\alpha }+N_{0}$$
(5)$$I_{k}=P_{i}h_{i,s}d_{i,s}^{-\alpha }+\sum_{j=1}^{N_{D}}P_{j}h_{j,s}d_{j,s}^{-\alpha }+\;\sum_{k^{\prime }=1,k^{\prime }\neq k}^{N_{S}}P_{k^{\prime }}h_{k^{\prime },s}d_{k^{\prime },s}^{-\alpha }+N_{0}$$
where $I_{i}$ is the combined interference received by eNB, $I_{j}$ is the same for the *j*th D2D receiver other than the *j*th transmitter, and $I_{k}$ is the same for all SUEs except *k*th to the SCeNB.

### 3.2. SINR Model

In wireless communication, the SNIR is measured as the ratio of the received power by the receiver to the total interference including spectral noise density. Since communications may take place in any of the previously mentioned three cases, the SINR at the *j*th D2D receiver is given by:(6)$$\gamma_{j}^{D}=\frac{P_{j}h_{t,r}d_{t,r}^{-\alpha }}{I_{j}}=\frac{P_{j}h_{t,r}d_{t,r}^{-\alpha }}{P_{i}h_{i,r}d_{i,r}^{-\alpha }+\;\sum_{j^{\prime }=1,j^{\prime }\neq j}^{N_{D}}P_{j^{\prime }}h_{j^{\prime },r}d_{j^{\prime },r}^{-\alpha }+\sum_{k=1}^{N_{S}}P_{k}h_{k,r}d_{k,r}^{-\alpha }+N_{0}}$$

Thus, according to Equation (2), the outage probability of the *j*th D2D link can be given as
$$P_{out,j}^{D}=P_{r}\left\{\gamma_{j}^{D}<\gamma_{th}\right\}=1-P_{r}\left\{\gamma_{j}^{D}\geq \gamma_{th}\right\}$$
where $\gamma_{th}$ is the required SINR threshold at the receiver for effective D2D communication.
$$P_{outj}^{D}=1-P_{r}\left\{\frac{P_{j}h_{t,r}d_{t,r}^{-\alpha }}{P_{i}h_{i,r}d_{i,r}^{-\alpha }+\;\sum_{j^{\prime }=1,j^{\prime }\neq j}^{N_{D}}P_{j^{\prime }}h_{j^{\prime },r}d_{j^{\prime },r}^{-\alpha }+\sum_{k=1}^{N_{S}}P_{k}h_{k,e}d_{k,e}^{-\alpha }+N_{0}}\geq \gamma_{th}\right\}$$
$$=1-P_{r}\left\{h_{t,r}\geq \frac{\gamma_{th}d_{t,r}^{\alpha }}{P_{j}}\left(P_{i}h_{i,r}d_{i,r}^{-\alpha }+\;\sum_{j^{\prime }=1,j^{\prime }\neq j}^{N_{D}}P_{j^{\prime }}h_{j^{\prime },r}d_{j^{\prime },r}^{-\alpha }+\sum_{k=1}^{N_{S}}P_{k}h_{k,r}d_{k,r}^{-\alpha }+N_{0}\right)\right\}$$

Since the channel coefficient is exponentially distributed, the expectation of interference from the above equation can be expressed as follows
(7)$$P_{outj}^{D}=1-E\left[exp\left(-\frac{\gamma_{th}d_{t,r}^{\alpha }}{P_{j}}\left(P_{i}h_{i,r}d_{i,r}^{-\alpha }+\;\sum_{j^{\prime }=1,j^{\prime }\neq j}^{N_{D}}P_{j^{\prime }}h_{j^{\prime },r}d_{j^{\prime },r}^{-\alpha }+\sum_{k=1}^{N_{S}}P_{k}h_{k,r}d_{k,r}^{-\alpha }+N_{0}\right)\right)\right]$$
where E(.) is the expectation function and $P_{outj}^{D}$ is the outage probability of *j*th DUEs with $\gamma_{th}$ SNIR threshold.

Here, channel quality follows the Rayleigh fading assumption, which is an exponentially distributed random variable. Assume $z=\frac{\gamma_{th}d_{t,r}^{\alpha }}{P_{j}}$ and $L_{I_{d}}\left(z\right)$ and $L_{I_{s}}\left(z\right)$ are the Laplace transformation of random variables $I_{d}$ and $I_{s}$ evaluated at z, respectively. Interference due to same cellular resource reuses by other D2D pairs and small cell links $I_{d}$ and $I_{s}$ are defined as $I_{d}=\;\sum_{j^{\prime }=1,j^{\prime }\neq j}^{N_{D}}P_{j^{\prime }}h_{j^{\prime },r}d_{j^{\prime },r}^{-\alpha }$ and $I_{s}=\;\sum_{k=1}^{N_{S}}P_{k}h_{k,r}d_{k,r}^{-\alpha }$, respectively. Therefore, Equation (7) can be written as
$$P_{out,j}^{D}=1-exp\left(-zN_{0}\right)L_{I_{d}}\left(z\right)LI_{s}\left(z\right)$$
$$=1-exp\left(\frac{-N_{0}\gamma_{th}d_{t,r}^{\alpha })}{P_{j}}\right)\left(\frac{P_{j,r}d_{i,r}^{\alpha }}{P_{i}\gamma_{th}d_{t,r}^{\alpha }+P_{j}d_{i,r}^{\alpha }}\right)LI_{d}\left(\frac{\gamma_{th}d_{t,r}^{\alpha }}{P_{j}}\right)LI_{s}\left(\frac{\gamma_{th}d_{t,r}^{\alpha }}{P_{j}}\right)$$
$$\begin{array}{c} =1-exp\left(\frac{-N_{0}\gamma_{th}d_{t,r}^{\alpha })}{P_{j}}\right)\left(\frac{P_{j}d_{i,r}^{\alpha }}{P_{i}\gamma_{th}d_{t,r}^{\alpha }+P_{j}d_{i,r}^{\alpha }}\right)exp\left(\frac{-\kappa {P_{j^{\prime }}}^{m}{\gamma_{th}}^{m}d_{t,r}^{2}\lambda_{d}}{{P_{j}}^{m}}\right)\\ exp\left(\frac{-\kappa {P_{k}}^{m}{\gamma_{th}}^{m}d_{t,r}^{2}\lambda_{s}}{{P_{j}}^{m}}\right) \end{array}$$
$$\begin{array}{c} =1-exp\left(\frac{-N_{0}\gamma_{th}d_{t,r}^{\alpha })}{P_{j}}\right)\left(\frac{P_{j}d_{i,r}^{\alpha }}{P_{i}\gamma_{th}d_{t,r}^{\alpha }+P_{j}d_{i,r}^{\alpha }}\right)exp\left(\frac{-\kappa {\gamma_{th}}^{m}d_{t,r}^{2}\left({P_{j^{\prime }}}^{m}\lambda_{d}+{P_{k}}^{m}\lambda_{s}\right)}{{P_{j}}^{m}}\right) \end{array}$$
(8)$$=1-\delta_{D}exp\left(-\beta_{D}\left({P_{j^{\prime }}}^{m}\lambda_{d}+{P_{k}}^{m}\lambda_{s}\right)\right)$$
where E(.) is the expectation function, $\delta_{D}=exp\left(\frac{-N_{0}\gamma_{th}d_{t,r}^{\alpha })}{P_{j}}\right)\left(\frac{P_{j}d_{i,r}^{\alpha }}{P_{i}\gamma_{th}d_{t,r}^{\alpha }+P_{j}d_{i,r}^{\alpha }}\right)$, κ=πmΓ(m)Γ(1−m), $\beta_{D}=\kappa {\gamma_{th}}^{m}d_{t,r}^{2}/{P_{j}}^{m}$, $\lambda_{d}$ is the density of DUEs for D2D pairs, and $\lambda_{s}$ is the density of small cells. The proof of the above equation can be referred to in Appendix A of [28].

From the above expression, it is clearly visible that the outage probability of D2D links depends on various factors such as path loss coefficient, required SINR, distances between UEs, transmission powers, and small cell and D2D pair density. Outage probability increases with the required SINR, but decreases when the distance between SUEs and the D2D receiver is increased.

Similarly, in the case of the macro-cellular communications mode where UE is served by eNB, the SINR at the receiver i can be expressed as:$$\gamma_{i}^{M}=\frac{P_{i}h_{i,e}d_{i,e}^{-\alpha }}{I_{i}}$$

Hence, the outage probability of the macro-cell link can be written as
(9)$$P_{out,i}^{M}=P_{r}\left\{\gamma_{i}^{M}<\gamma_{th}\right\}=1-P_{r}\left\{\gamma_{i}^{M}\geq \gamma_{th}\right\}$$
where $\gamma_{th}$ is the required SINR for the *i*th CUE for effective cellular communication. After substituting the SINR values, the outage probability of the macro-cellular link will be as follows
$$P_{out,i}^{M}=1-P_{r}\left\{\frac{P_{i}h_{i,e}d_{i,e}^{-\alpha }}{\sum_{j=1}^{N_{D}}P_{j}h_{j,e}d_{j,e}^{-\alpha }+\sum_{k=1}^{N_{S}}P_{k}h_{k,e}d_{k,e}^{-\alpha }+N_{0}}\geq \gamma_{th}\right\}$$
$$=1-P_{r}\left\{h_{i,e}\geq \frac{\gamma_{th}d_{i,e}^{\alpha }}{P_{i}}\left(\sum_{j=1}^{N_{D}}P_{j}h_{j,e}d_{j,e}^{-\alpha }+\sum_{k=1}^{N_{S}}P_{k}h_{k,e}d_{k,e}^{-\alpha }+N_{0}\right)\right\}$$
$$=1-E\left[exp\left(-\frac{\gamma_{th}d_{i,e}^{\alpha }}{P_{i}}\left(\sum_{j=1}^{N_{D}}P_{j}h_{j,e}d_{j,e}^{-\alpha }+\sum_{k=1}^{N_{S}}P_{k}h_{k,e}d_{k,e}^{-\alpha }+N_{0}\right)\right)\right]$$
$$=1-exp\left(-\frac{N_{0}\gamma_{th}d_{i,e}^{\alpha }}{P_{i}}\right)L_{I_{d}}\left(\frac{\gamma_{th}d_{i,e}^{\alpha }}{P_{i}}\right)L_{I_{s}}\left(\frac{\gamma_{th}d_{i,e}^{\alpha }}{P_{i}}\right)$$
$$=1-exp\left(-\frac{N_{0}\gamma_{th}d_{i,e}^{\alpha }}{P_{i}}\right)exp\left(-\frac{\kappa {\gamma_{th}}^{m}{d_{i,e}}^{2}\left(P_{j}^{m}\lambda_{d}+P_{k}^{m}\lambda_{s}\right)}{P_{i}^{m}}\right)$$
(10)$$=1-\delta_{M}exp\left(-\beta_{M}\left({P_{j}}^{m}\lambda_{d}+{P_{k}}^{m}\lambda_{s}\right)\right)$$
where $\delta_{M}=exp\left(-\frac{N_{0}\gamma_{th}^{m}d_{i,e}^{\alpha }}{P_{i}}\right)$ and $\beta_{M}=\frac{\kappa {\gamma_{th}}^{m}d_{i,e}^{2}}{{P_{i}}^{m}}$. Hence, the outage probability decreases with increasing distances between SUEs and UEs, as well as UEs and D2D transmitters.

For small cell mode communications, the UE is served by the small cell SCeNB, and the link between the SUE and the SCeNB will be interfered with by other SCeNBs, D2D pairs, and macro-eNB. Therefore, the SINR at the small cell receiver can be written as
$${\gamma_{k}}^{S}=\left(P_{k}h_{k,s}d_{k,k}^{-\alpha }\right)/I_{k}$$
and the outage probability of the small cell link can be expressed as
(11)$$P_{out,k}^{S}=P_{r}\left\{\gamma_{i}^{S}<\gamma_{th}\right\}=1-P_{r}\left\{\gamma_{i}^{S}\geq \gamma_{th}\right\}$$

Thus, after substituting all the values, the outage probability of small cell cellular link can be derived as
$$P_{out,k}^{S}=1-P_{r}\left\{\frac{P_{k}h_{k,s}d_{k,s}^{-\alpha }}{P_{i}h_{i,s}d_{i,s}^{-\alpha }+\sum_{j=1}^{N_{D}}P_{j}h_{j,s}d_{j,s}^{-\alpha }+\;\sum_{k^{\prime }=1,k^{\prime }\neq k}^{N_{S}}P_{k^{\prime }}h_{k^{\prime },s}d_{k^{\prime },s}^{-\alpha }+N_{0}}\geq \gamma_{th}\right\}$$
$$=1-P_{r}\left\{h_{k,s}\geq \frac{\gamma_{th}d_{k,s}^{\alpha }}{P_{k}}\left(P_{i}h_{i,s}d_{i,s}^{-\alpha }+\sum_{j=1}^{N_{D}}P_{j}h_{j,s}d_{j,s}^{-\alpha }+\;\sum_{k^{\prime }=1,k^{\prime }\neq k}^{N_{S}}P_{k^{\prime }}h_{k^{\prime },s}d_{k^{\prime },s}^{-\alpha }+N_{0}\right)\right\}$$
$$=1-E\left[exp\left(-\frac{\gamma_{th}d_{k,s}^{\alpha }}{P_{k}}\left(P_{i}h_{i,s}d_{i,s}^{-\alpha }+\sum_{j=1}^{N_{D}}P_{j}h_{j,s}d_{j,s}^{-\alpha }+\;\sum_{k^{\prime }=1,k^{\prime }\neq k}^{N_{S}}P_{k^{\prime }}h_{k^{\prime },s}d_{k^{\prime },s}^{-\alpha }+N_{0}\right)\right)\right]$$
$$=1-exp\left(-\frac{N_{0}\gamma_{th}d_{k,s}^{\alpha }}{P_{k}}\right)\left(\frac{P_{k}{d_{k,e}}^{\alpha }}{P_{i}\gamma_{th}d_{k,s}^{\alpha }+P_{k}{d_{k,e}}^{\alpha }}\right)L_{I_{d}}\left(\frac{\gamma_{th}d_{k,s}^{\alpha }}{P_{k}}\right)L_{I_{s^{\prime }}}\left(\frac{\gamma_{th}d_{k,s}^{\alpha }}{P_{k}}\right)$$
$$=1-\left(\frac{P_{k}d_{k,e}^{\alpha }}{P_{i}\gamma_{th}{d_{k,s}}^{\alpha }+P_{k}{d_{k,e}}^{\alpha }}\right)exp\left(-\frac{N_{0}\gamma_{th}d_{k,s}^{\alpha }}{P_{k}}\right)exp\left(-\frac{\kappa {\gamma_{th}}^{m}{d_{k,s}}^{2}\left(P_{j}^{m}\lambda_{d}+P_{k^{\prime }}^{m}\lambda_{s}\right)}{P_{k}^{m}}\right)$$
(12)$$=1-\delta_{S}exp\left(-\beta_{S}\left({P_{j}}^{m}\lambda_{d}+{P_{k^{\prime }}}^{m}\lambda_{s}\right)\right)$$
where $\delta_{S}=\left(\frac{P_{k}d_{k,e}^{\alpha }}{P_{i}\gamma_{th}{d_{k,s}}^{\alpha }+P_{k}{d_{k,e}}^{\alpha }}\right)exp\left(-\frac{N_{0}\gamma_{th}d_{k,s}^{\alpha }}{P_{k}}\right)$, $\beta_{S}=\frac{\kappa {\gamma_{th}}^{m}d_{k,s}^{2}}{{P_{k}}^{m}}$, and $I_{s^{\prime }}$ is the interference from all SCeNBs except respective small cell links.

According to the above expressions of outage probabilities, it is easily visible that intensity of interference (i.e., the probability of success) depends on the density of small cells and D2D pairs, the distances between the receiver and transmitter of those links, the required SINR threshold, and the transmission power. Increasing the DUE receiver distance from CUEs or SUEs will increase the probability of success for D2D links. Similarly, increasing the distance between the D2D transmitter and CUEs or SUEs will decrease the outage probability of SceNB links.

Based on the previous expressions, we propose Algorithm 1 (see below), called acceptable interference regions (AIR), in which link QoS for various communication modes will be guaranteed by limiting the coexistence of different UEs under D2D link constraints as follows:(13a)$$d_{ij}^{D}\geq d_{ij}^{D\;\;min}$$
(13b)$$d_{je}^{M}\geq d_{je}^{M\;\;min}$$
(13c)$$d_{js}^{S}\geq d_{ij}^{S\;\;min}$$
where $d^{min}$ is the lower boundary that must be satisfied to maintain the QoS for various links; it can be measured as shown in Appendix A. For any communications, UEs will fulfill the minimum distance requirement to guarantee the link QoS.
**Algorithm 1:** The proposed AIR dynamic algorithm.
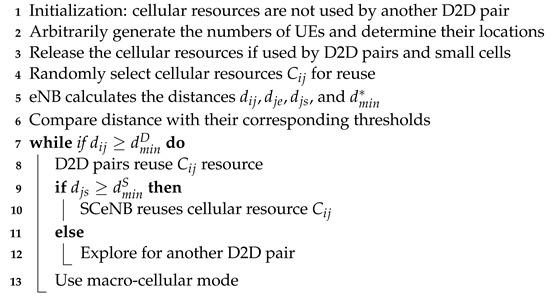


### 3.3. Small Cell Density Model

In Figure 2, both small cells and UEs are considered to be distributed based on separate Poisson point processes.

Previous analysis indicates that mutual interference is heavily dependent on small cell and D2D pairs densities. Small cell density, $\gamma_{s}$ beyond a certain threshold causes excessive interference for D2D communications, which resulted in no solution for the AIR feasible set, and one or more links will fail to satisfy QoS requirements. Contrarily, lower values of $\gamma_{s}$ yields higher feasible DUEs. Nevertheless, lower $\gamma_{s}$ may result in a smaller overall transmission rate for small cell UEs. In general, the transmission rate of small cell UEs is defined as the number of successful transmissions per unit area [29]. Therefore, the transmission capacity of small cells can be expressed as
(14)$$T_{c}=\lambda_{s}\left(1-P_{o}ut^{S}\right)=\lambda_{s}\delta_{S}exp\left(-\beta_{S}\left({P_{j}}^{m}\lambda_{d}+{P_{k^{\prime }}}^{m}\lambda_{s}\right)\right)$$

An optimum problem for the above scenario can be formulated as
(15)$$Max\;\;T_{c}=\lambda_{s}\delta_{S}exp\left(-\beta_{S}\left({P_{j}}^{m}\lambda_{d}+{P_{k^{\prime }}}^{m}\lambda_{s}\right)\right)$$

An optimum value for small cell density, $\gamma_{s}$ can be obtained by maximizing Equation (17) for small cells with or without satisfying the QoS requirements for the various links. Thus, the optimum solution for the above equation without considering QoS constraints is as follows:$$\tilde{\lambda_{s}}=\frac{1}{P_{k\prime }^{m}\beta_{S}}=\frac{P_{k}^{m}}{P_{k\prime }^{m}\gamma_{th}d_{k,s}^{2}}$$

Therefore, under fixed transmission power conditions, a higher number of SCeNBs can be added into the network for sharing cellular resources by reducing either the distance between the SCeNB and the SUE or the required SINR threshold. The proof of the solution for this optimal problem is shown in Appendix B.

Similarly, for D2D pairs density, we can obtain
$$\tilde{\lambda_{d}}=\frac{1}{P_{j\prime }^{m}\beta_{D}}=\frac{P_{j}^{m}}{P_{j\prime }^{m}\gamma_{th}d_{t,r}^{2}}$$

As known from previous analysis of outage probabilities for DUEs, CUEs, and SUEs, it is very hard to avoid the monotonically increasing nature of success probabilities with increasing small cell density $\lambda_{s}$, or D2D pairs density $\lambda_{d}$, or both. However, increasing $\lambda_{s}$ or $\lambda_{d}$ introduces additional interference. Hence, to obtain an optimum value for small cell density $\lambda_{s}$ by fulfilling QoS requirements for all communications mode, Equation (18) must satisfy the following constraints:(16a)$$P_{out}^{D}\leq \tau$$
(16b)$$P_{out}^{M}\leq \tau$$
(16c)$$P_{out}^{S}\leq \tau$$
where τ is the maximum allowable outage probabilities for any links. As controlling the number of small cells is easier compared to controlling DUEs, according to the above constraints in Equation (16), the density of small cells must satisfy the following:(17)$$\lambda_{s}\leq min\left\{f\left(\tau \right),g\left(\tau \right),h\left(\tau \right)\right\}$$
where
$$\begin{array}{c} f\left(\tau \right)={argmax}_{\lambda_{s}>0}\left\{P_{out,j}^{D}\leq \tau \right\}\\ g\left(\tau \right)={argmax}_{\lambda_{s}>0}\left\{P_{out,i}^{M}\leq \tau \right\}\\ h\left(\tau \right)={argmax}_{\lambda_{s}>0}\left\{P_{out,k}^{S}\leq \tau \right\} \end{array}$$

Hence, by considering QoS constraints in Equation (16a–c), we can obtain the solution for small cell density $\lambda_{s}$ for various communications modes as follows:$$\begin{array}{c} \lambda_{s}^{S}\leq \frac{1}{\beta_{S}P_{k\prime }^{m}}ln\frac{\delta_{S}}{1-\tau }-\rho_{jk\prime }^{m}\lambda_{d}\\ \lambda_{s}^{M}\leq \frac{1}{\beta_{M}P_{k}^{m}}\ln\frac{\delta_{M}}{1-\tau }-\rho_{j\prime k}^{m}\lambda_{d}\\ \lambda_{s}^{D}\leq \frac{1}{\beta_{D}P_{k}^{m}}ln\frac{\delta_{D}}{1-\tau }-\rho_{jk}^{m}\lambda_{d} \end{array}$$

The proof for this solution is shown in Appendix C. Here, $\lambda_{s}^{S}$, $\lambda_{s}^{S}$, and $\lambda_{s}^{S}$ are the small cell densities for small cell, macro-cell, and D2D mode communications, respectively, and ρ is the ratio of transmission powers. Hence, the arguments of Equation (17) are as follows:$$\begin{array}{c} f\left(\tau \right)=\frac{1}{\beta_{S}P_{k\prime }^{m}}\ln\left(\frac{\delta_{S}}{1-\tau }\right)-\rho_{jk\prime }^{m}\lambda_{d}\\ g\left(\tau \right)=\frac{1}{\beta_{M}P_{k}^{m}}\ln\left(\frac{\delta_{M}}{1-\tau }\right)-\rho_{j\prime k}^{m}\lambda_{d}\\ h\left(\tau \right)=\frac{1}{\beta_{D}P_{k}^{m}}\ln\left(\frac{\delta_{D}}{1-\tau }\right)-\rho_{jk}^{m}\lambda_{d} \end{array}$$

Therefore, the outage probability constraints of the optimal problem in Equation (17) can be represented with constraints of small cell density as follows:$$\begin{array}{c} \lambda_{s}\leq min\left\{\left\{\frac{1}{\beta_{S}P_{k\prime }^{m}}\ln\left(\frac{\delta_{S}}{1-\tau }\right)-\rho_{jk\prime }^{m}\lambda_{d}\right\},\left\{\frac{1}{\beta_{M}P_{k}^{m}}\ln\left(\frac{\delta_{M}}{1-\tau }\right)-\rho_{j\prime k}^{m}\lambda_{d}\right\},\left\{\frac{1}{\beta_{D}P_{k}^{m}}\ln\left(\frac{\delta_{D}}{1-\tau }\right)-\rho_{jk}^{m}\lambda_{d}\right\}\right\}=\lambda_{s}^{Max} \end{array}$$
where $\lambda_{s}^{Max}$ is the maximum allowable small cell density to guarantee the links QoS. Hence, to maximize the transmission capacity of small cells, we can propose a transmission ON-OFF Algorithm 2 for small cells.
**Algorithm 2.** ON-OFF algorithm for interference minimization.
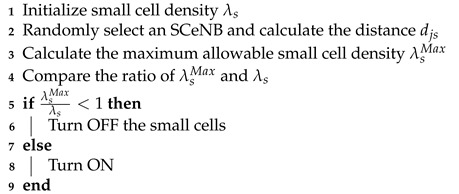


## 4. Performance Evaluation

### 4.1. Simulation Environment and Parameters

The system model was validated using a MATLAB-based simulation. Table 2 lists the parameters used in the simulation for the D2D-enabled cellular HetNets. In our simulation setup, we considered a single eNB with a 500 m cell radius located at the center of D2D-enabled HetCNets where CUEs are randomly distributed. DUEs and SUEs were realized according to two independents PPPs with densities of $\lambda_{d}$ and $\lambda_{s}$, respectively. The number of CUEs was selected in such a way that the saturation condition is always satisfied. Our analytical model is only valid under the assumption that each eNB has at least one user to serve in the uplink. We evaluated the coverage probability of the proposed scheme with an average of 10,000 independent realizations.

### 4.2. Results and Discussion

The proposed model is based on the distance-based mode selection strategy without power control. As DUEs are distributed based on the PPP function, higher distances between the D2D receiver and transmitter are more likely to occur, which eventually increases the transmission power and associated interference. Hence, the proposed scheme with an added power control mechanism can be part of our future work.

Due to the limited related published work in the field, the proposed AIR algorithm and the resulting scheme was validated by comparing it with the traditional neighbor-based scheme, in which information is transmitted via D2D transmission to the targeted neighboring UEs, as well as the work presented in [12], in which D2D transmission reuses resources only if the D2D links satisfy the given QoS requirements with guaranteed CUEs transmission. Initially, the cumulative distribution functions (CDFs) of outage probabilities of all three communications modes were evaluated and compared with the above-mentioned baseline schemes.

In Figure 3, we plot CDF versus the outage probability of the D2D link. We observe that the outage probability of the D2D link was improved by up to 55% using the proposed scheme when compared with the neighbor-based scheme. However, the improvement is marginal (up to 3%) when contrasted with the scheme proposed in [12], where the interference is considered for single SCeNB and D2D pairs. It is also clearly visible that the outage probability decreases with the decrease of τ (QoS requirements for D2D links); more than 88% of the D2D links can meet the QoS requirements in this scenario.

In Figure 4 and Figure 5, macro-cell and small cell outage probabilities are compared with the same baseline. Similar to the D2D link outage case, outage probability improvement is noticeable in our proposed scheme. However, the improvements are not as prominent as those in D2D links; the reason behind this is that D2D transmitters are chosen based on the AIR scheme, which is designed to minimizing the D2D interference. In addition, due to smaller path loss, the channel quality of D2D links is much better compared with the macro-cell and small cell links. Despite all these factors, here we can see that at CDF = 0.7, the outage probability of the feasible set AIR scheme is much better when compared to the neighbor-based scheme (35% in eNB and 69% in SCeNB links) and is considerably better when compared to the scheme in [12] (6% in eNB and 8% in SCeNB links).

From Figure 6, we observe that the outage probability increases with an increase in SINR threshold requirements and at lower SINR threshold values, the cellular coverage is nearly perfect; i.e., cellular outage is almost zero. By controlling interference, we can minimize the required threshold for decoding the message and hence improve the performance in terms of link availability. Figure 7 and Figure 8 also reveal the impact of densities of D2D pairs and small cells on the link availability. In both cases, outage probability increases with increasing D2D pairs and small cell densities.

In Figure 9, transmission capacity increases with the path loss coefficient α due to the increase in the fading of interference. By looking at Figure 9, we observe that for α=4, a similar transmission capacity is almost achieved with the AIR scheme at 4.1 $\times \;10^{-5}$ for optimal small cell density, compared to 5.31 $\times \;10^{-5}$ for the same in the scheme presented in [12]. For α=3, an equivalent transmission capacity is achieved for optimal small cell densities of 3.7 $\times \;10^{-5}$ and 4.23 $\times \;10^{-5}$ in AIR and the scheme of [12], respectively.

## 5. Conclusions

In this paper, we developed mathematical models of networks, SNIR, and small cell density for system performance modeling and analysis. In addition, we derived the outage probabilities of D2D, macro-cells, and small cell links. Finally, we proposed a dynamic algorithm called acceptance interference region (AIR) to provide a solution to the problem of achieving a strict QoS guarantee to all links in D2D-enabled HetCNets. Our analytical models were validated using a MATLAB-based simulation. The simulation results show that the proposed AIR scheme achieved an improved outage probability of 35% and 49% in eNB and SCeNB links, respectively, when compared with the traditional neighbor-based methods. We also proposed an efficient ON/OFF algorithm to achieve better transmission capacity in the network than with existing methods. We found that the transmission capacity is maximized at lower small cell densities. Developing a test-bed measurement system to further validate the system performance is suggested as future research work.

## Figures and Tables

**Figure 1 sensors-22-01063-f001:**
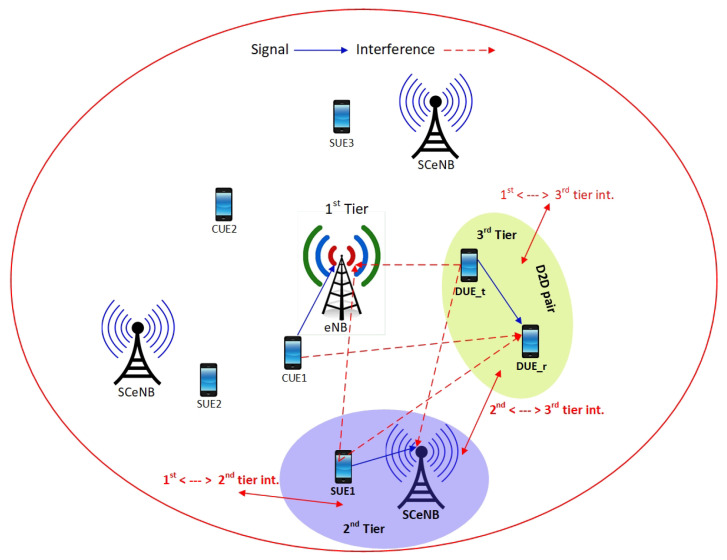
Interference scenario in a D2D-enabled three-tier cellular network.

**Figure 2 sensors-22-01063-f002:**
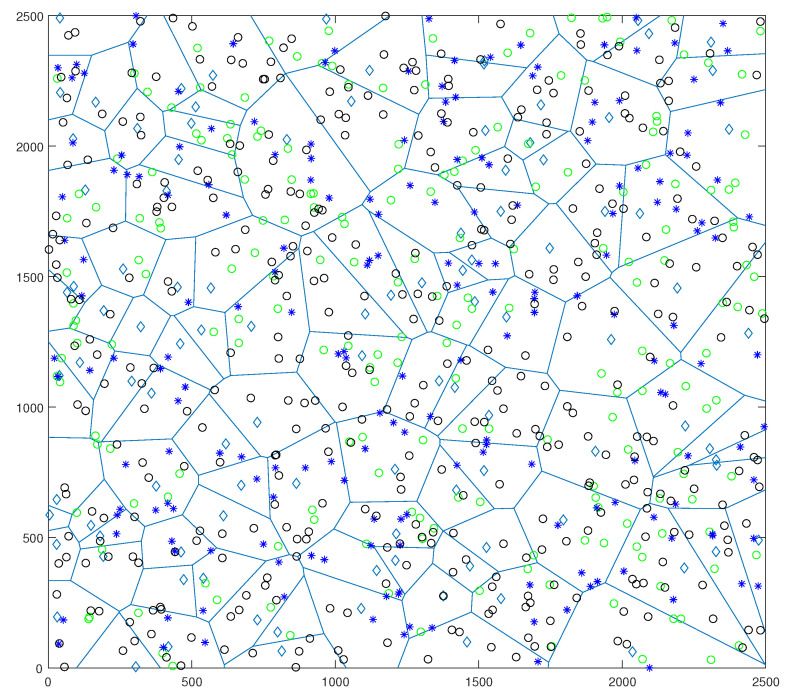
PPP distributions of Nodes. Diamonds represent the eNB, circles represent UEs (green circles are DUEs), and asterisks represent SCeNBs in a cellular network.

**Figure 3 sensors-22-01063-f003:**
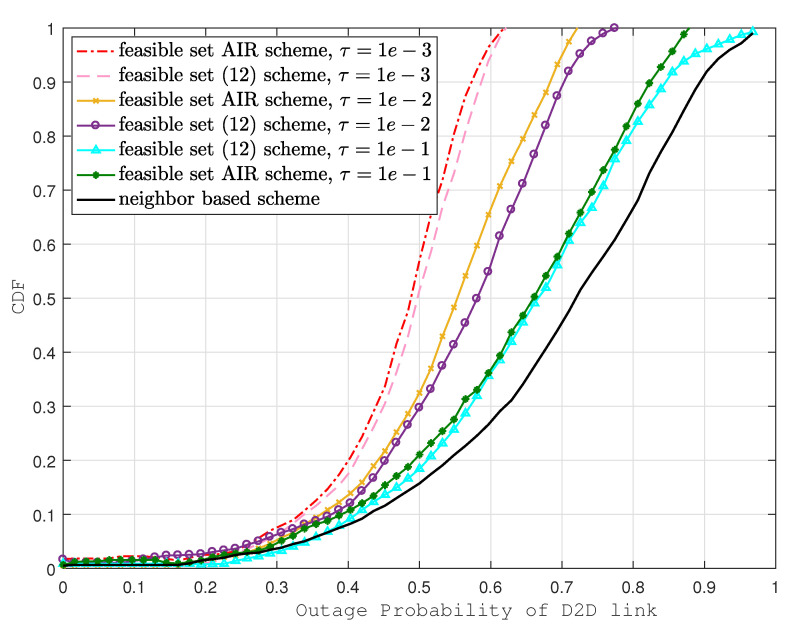
Outage probability of D2D links with different schemes.

**Figure 4 sensors-22-01063-f004:**
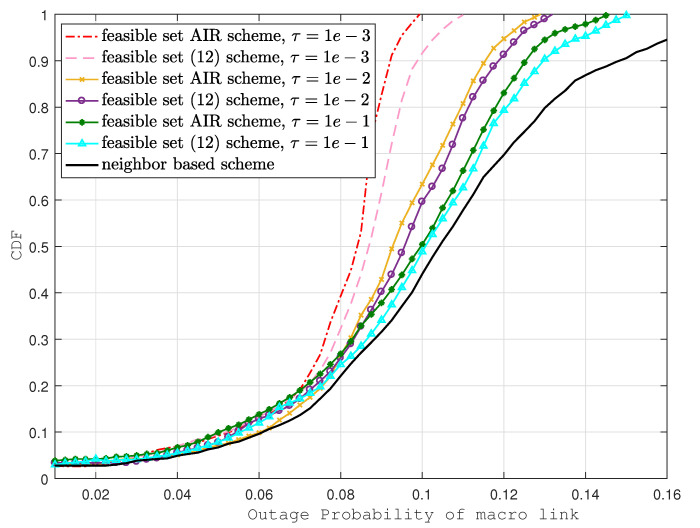
Outage probability of macro-links with various schemes.

**Figure 5 sensors-22-01063-f005:**
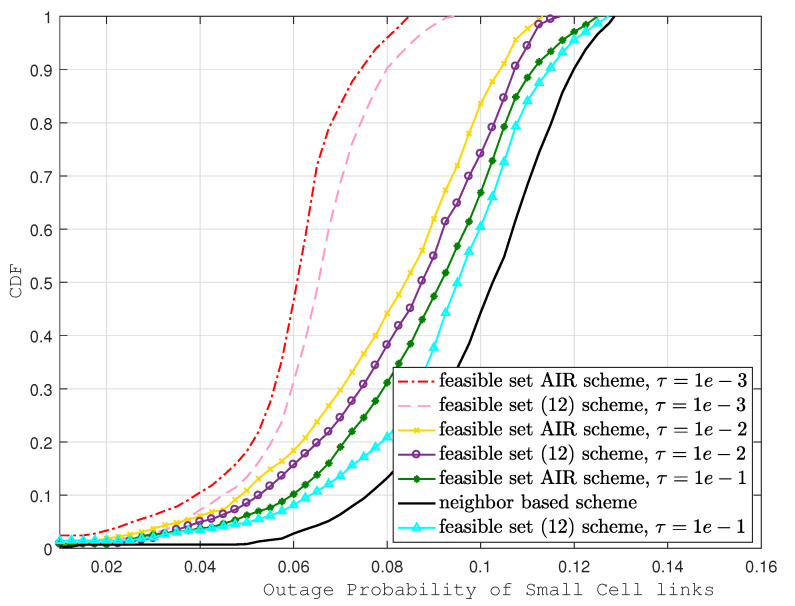
Outage probability of small cell links with various schemes.

**Figure 6 sensors-22-01063-f006:**
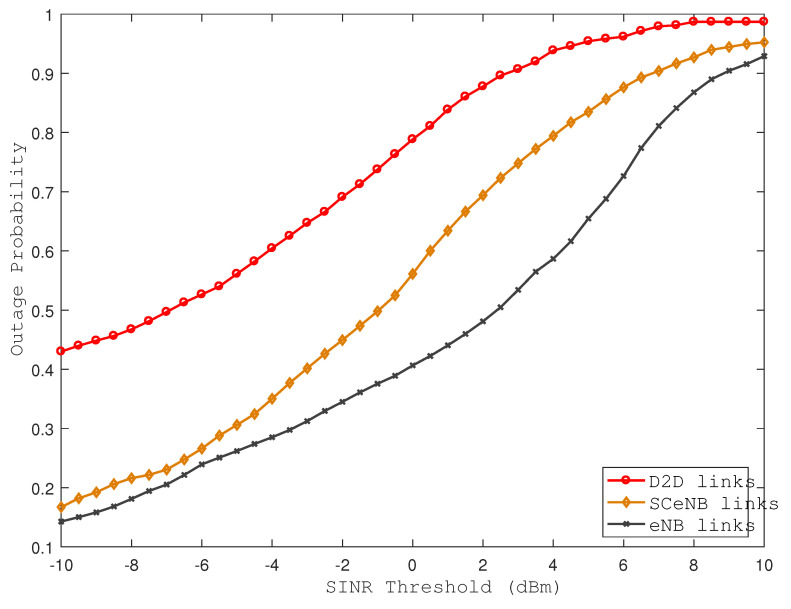
Outage probability of different links with various SINR thresholds.

**Figure 7 sensors-22-01063-f007:**
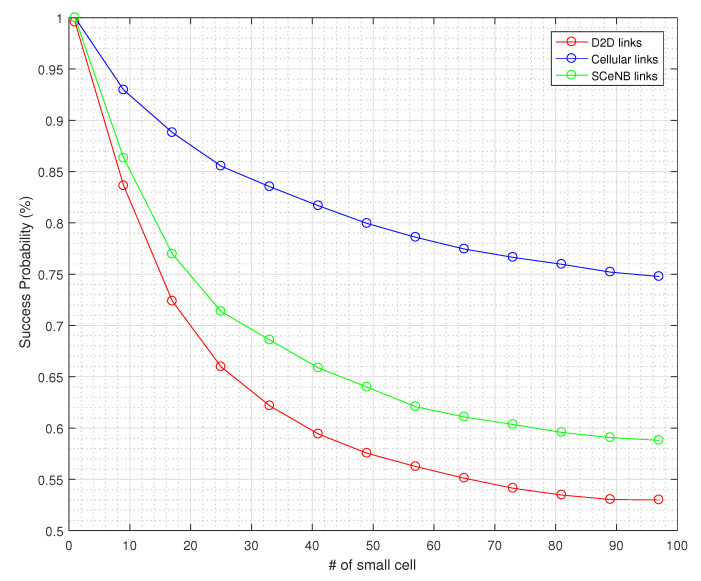
Success probability with small cell density.

**Figure 8 sensors-22-01063-f008:**
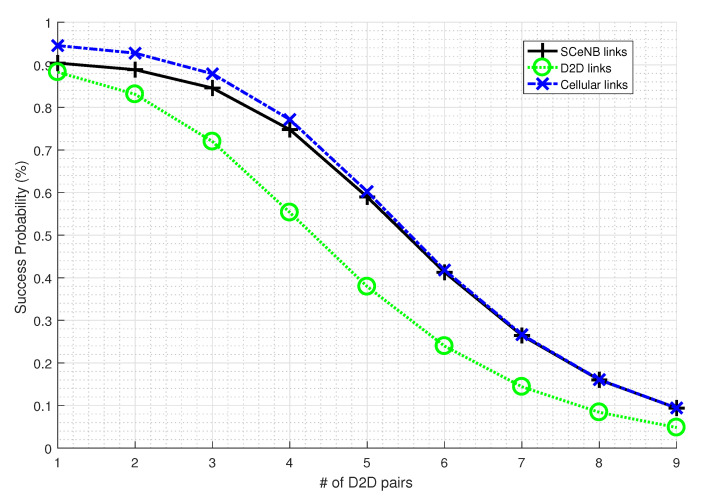
Success probability with D2D pair density.

**Figure 9 sensors-22-01063-f009:**
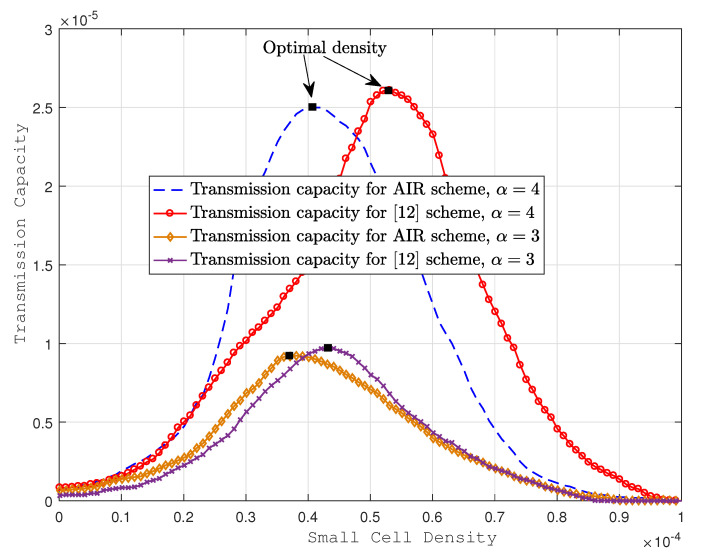
Transmission capacities for optimal deployment of various schemes.

**Table 1 sensors-22-01063-t001:** List of key notations and abbreviations used.

Notation	Definition
Φ	PPP constituted by the macro-BSs (eNB)
$\Phi_{s}$	PPP constituted by the small cell BSs (SCeNB)
$\Phi_{d}$	PPP constituted by DUEs
$\lambda_{s}$	Intensity of SCeNBs
$\lambda_{d}$	Intensity of DUES
$P_{c}$	Transmission power of a typical UE operating in cellular mode
$P_{d}$	Transmission power of a typical UE operating in D2D mode
$P_{s}$	Transmission power of a typical SUE
$N_{C},N_{D},N_{S}$	The set of CUEs, DUEs, and SUEs
α	Path loss exponent
$\gamma_{th}$	Required SINR for cellular and D2D links
$N_{0}$	Thermal noise
$h_{x,y}$	Channel coefficient between x and y
D2D	Device-to-device
SINR	Signal-to-noise-plus-interference ratio
UE	User equipment
CUE	Cellular user equipment
DUE	Device-to-device user equipment
MS	Mode selection
PPP	Poison point process
SUE	Small cell user equipment
eNB	Evolved node B, i.e., LTE macro-base station
SCeNB	Small cell evolved node B, i.e., small cell
AIR	Accepted interference region
QoS	Quality of Service

**Table 2 sensors-22-01063-t002:** Parameters used in the simulation.

Simulation Parameters	Values
Intensity of SCeNBs, $\lambda_{s}$	$10^{-5}$
Intensity of DUES, $\lambda_{d}$	$10^{-3}$
CUE transmission power, $P_{i}$	23 (dBm)
Transmission power of DUEs, $P_{j}$	20 (dBm)
Transmission power of SUEs, $P_{k}$	20 (dBm)
Path loss exponent, α	3 and 4
Required SINR threshold, $\gamma_{th}$	−2.6 dB
Noise power, $N_{0}$	−118 dBm
Cellular UE numbers, $N_{C}$	150
D2D pairs numbers, $N_{D}$	50
Small cell UE numbers, $N_{S}$	250
Maximum allowable outages, i.e., QoS, τ	0.1, 0.01 and 0.001
Channel bandwidth, B	180 KHz

## Data Availability

Not applicable.

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
