# Peer review of "A Dynamic Algorithm for Interference Management in D2D-Enabled Heterogeneous Cellular Networks: Modeling and Analysis"

_sensors, 2022, doi:10.3390/s22031063_

Round 1

Reviewer 1 Report

In this paper, an algorithm for improving the SINR in D2D enabled heterogeneous cellular network has been presented. The paper represents an extension of the results presented in [12] and could be found helpful for people working in this area. However, a few points should be clarified before it can be accepted

  1. The reference list should be updated by including relevant and recent papers in this topic [R-1]
  2. Page 6/20, please correct poison point process as well as “In a 3-tier networks”
  3. In (3) define d_{xy} and explain how is it related to D in (2)
  4. Define Pr in (4)
  5. It is not clear how (3) is related to (5) (6) (7). The authors should improve this part
  6. The definition of E(.) should be provided below (9)
  7. Above (9) what do you mean by “exponentially distributed (Poisson point distribution)”? It is exponential or Poisson distributed?
  8. Please revise “Here channel coefficient…variable with unit mean”
  9. \kappa defined below (14) has been defined several times in the manuscript
  10. PPP has been defined several times
  11. Please provide additional information for the traditional neighbour-based scheme and the one presented in [12]

[R-1] ~ "An SINR-aware joint mode selection, scheduling, and resource allocation scheme for D2D communications." IEEE Transactions on Vehicular Technology 68.5 (2019): 4949-4963.

[R-2] ~ "Energy-efficient mode selection for D2D communications in cellular networks." IEEE Transactions on Cognitive Communications and Networking 4.4 (2018): 869-882.

Reviewer 2 Report

This manuscript reviews the problems of MS as well as resource allocation for mutual interference in 3-tier cellular networks and presents a dynamic algorithm based on a distance-based approach to minimize the interference and to guarantee QoS for both cellular and D2D communication links for the network researchers. This manuscript provides relevant analysis to the audiences ONLY. However, the following should be carefully considered:

1- The entire text of the manuscript should be reviewed in English, MUST. The sluggish/repeated sentences can be omitted or careful write up can save the lines. Like first four and last three lines can be reduced to two and one-lines respectively. 

2-Similarly, the Conclusion is not up to the mark. Refer to the journal papers to redevelop/rewrite  

3- make sure figures are always on the top/bottom of the article, not inserted in the middle of the text.

4- Figure-Fig,figs ............................. Inconsistent

5. x and y labels in the figures can be made more clearer. Legends should be placed in the top left or changed orientation as in your case. Legends should be in the Figure box not outside.

6. Abbreviations must be mentioned in the abstract and paper once, at least.  CUE, DUE and many other

7- Basic or first equations be cited/given reference

8- The citation of the paper is poor. Limit references to 2016, possibly. and supplement your paper with the following papers

i.  Lightweight Elliptic-Elgamal-Based Authentication Scheme for Secure Device-to-Device Communication, https://doi.org/10.3390/fi11050108

ii. An Efficient Resource Allocation Algorithm for Device-To-Device Communications, https://doi.org/10.3390/app9183816

iii. Deep reinforcement learning-based resource allocation for D2D communications in heterogeneous cellular networks, https://doi.org/10.1016/j.dcan.2021.09.013

8. Mention the reference at the end of the text, once. see in the case of Reference 1 and 2, others

Round 2

Reviewer 1 Report

The authors have addressed the comments raised in the previous round of reviews